# T-WaveNet: A Tree-Structured Wavelet Neural Network for Time Series Signal Analysis

**Minhao LIU[†\*], Ailing Zeng[†], Qiuxia LAI[†], Ruiyuan Gao[†], Min Li[†], Jing Qin[‡], Qiang Xu[†\*]**
[†]Department of Computer Science & Engineering, The Chinese University of Hong Kong
[‡]School of Nursing, The Hong Kong Polytechnic University
\*{mhliu,qxu}@cse.cuhk.edu.hk

## Abstract

Time series signal analysis plays an essential role in many applications, e.g., activity recognition and healthcare monitoring. Recently, features extracted with deep neural networks (DNNs) have shown to be more effective than conventional hand-crafted ones. However, most existing solutions rely solely on the network to extract information carried in the raw signal, regardless of its inherent physical and statistical properties, leading to sub-optimal performance particularly under a limited amount of training data.

In this work, we propose a novel tree-structured wavelet neural network for time series signal analysis, namely *T-WaveNet*, taking advantage of an inherent property of various types of signals, known as the *dominant frequency range*. Specifically, with *T-WaveNet*, we first conduct frequency spectrum energy analysis of the signals to get a set of dominant frequency subbands. Then, we construct a tree-structured network that iteratively decomposes the input signal into various frequency subbands with similar energies. Each node on the tree is built with an invertible neural network (INN) based wavelet transform unit. Such a disentangled representation learning method facilitates a more effective extraction of the discriminative features, as demonstrated with the comprehensive experiments on various real-life time series classification datasets.

## 1 Introduction

Time-varying signal analysis plays a crucial role in various applications. For example, smartwatches utilize inertial signals for human activity logging (Khan et al., 2019); brain-computer interfaces employ electroencephalography (EEG) signals to identify user intentions (Zhang et al., 2020; Autthasan et al., 2021); clinical diagnosis systems use surface electromyography signals (sEMG) for neuromuscular pathological analysis (Duan et al., 2015), and develop muscle-computer interfaces to control external devices (Pancholi et al., 2021).

The signals mentioned above are all typical time series data (i.e., a set of observations collected and ordered chronologically) and express information at specific frequency ranges. Generally speaking, time series signal analysis consists of three steps: (i) *data segmentation*, wherein the continuous signals are partitioned into segments using fixed- or variable-sized windows; (ii) *feature extraction*, wherein various techniques are applied on each segment to extract distinguishing features; and (iii) *downstream tasks*, which generate the desired outputs for certain tasks (e.g., classification and anomaly detection) with the extracted features.

Among the three steps, feature extraction is usually the most critical one. Traditional feature extraction approaches for time series signals can be broadly categorized as *statistical* and *structural* methods (Lara et al., 2012). The former utilizes statistical measurements in time- or frequency-domain to figure out discriminative features, where exemplar time-domain measurements are mean and variance (Kao et al., 2009; Vepakomma et al., 2015), and typical frequency-domain measurements include short-time Fourier transform (STFT) and discrete Wavelet transform (DWT) (Jiang & Yin, 2015; Duan et al., 2015)). On the other hand, structural methods aim at describing the morphological interrelationship among the data via polynomial and/or exponential analysis (Olszewski, 2001).

While the above hand-crafted features are efficient for some simple signal analysis tasks, their limited representation capability makes them incapable of dealing with complicated signals contaminated by noise or artifacts, and hence less competitive in real-world tasks (Chen et al., 2021). Recently, deep neural networks (DNNs) have become the mainstream approaches for feature extraction in time series signal analysis. Various DNN models, including CNN-based solutions (Xi et al., 2018; Lawhern et al., 2018; Lin et al., 2020; Amin et al., 2019), CNN-LSTM combined models (Wang et al., 2020; Xu et al., 2019; Yuki et al., 2018), and Transformer-based techniques (Song et al., 2021; Li et al., 2021), are proposed in the literature, achieving promising performance in many tasks.

Despite of the remarkable success of these deep models, most of them place heavy demands on a large amount of labeled data. However, in many real-world applications, it is difficult and/or expensive to acquire sufficient labeled data to train these models. In addition, noise and artifacts that commonly exist in the time series signals make it even harder to obtain discriminative and robust representations. To the end, it may not fully reveal the superiority of deep models when solely relying on the network to directly extract information from raw signals. To solve this problem, some works attempt to use prior knowledge or hand-crafted features to guide the training of the deep learning models, aiming at extracting more effective features with a limited amount of training data (Ito et al., 2018; Wei et al., 2019; Laput & Harrison, 2019). However, these methods either simply leverage prior knowledge as gates to select features or intuitively add some statistical measurements as regularization terms to the losses, which fail to bring significant improvement. Moreover, designing tailored feature sets for different kinds of signals is tedious and time-consuming, which severely limits the generalization capability of such methods. In this regard, a promising way to acquire more discriminative features is to explore approaches to deeply integrating the inherent properties of the signals into the training process of the deep learning models.

An inherent property termed as the *dominant frequency range* has been evidenced in many time series signals, which is a small subset of the frequency components that carries the primary information of the signal (Telgarsky, 2013). For instance, more than $95\%$ of human body motion energy exists in the frequency components below 15 Hz (Karantonis et al., 2006); the informative frequency ranges in brain signals (EEG) have been discovered and named as $\delta$ (0.1-3 Hz) , $\theta$ (4-7 Hz), $\alpha$ (8–13Hz), $\beta$ (14-30 Hz) and $\gamma$ (31-100Hz) (CR & MP, 2011). However, most existing deep learning models are unaware of this essential property, and do not fully consider the different roles of various frequency components, leading to sub-optimal solutions. Motivated by the above, we propose a novel tree-structured wavelet neural network, namely *T-WaveNet*, to extract more effective features from time-series signals by seamlessly and effectively integrating this property into a deep model. Different from previous models, the proposed *T-WaveNet* adaptively represents the dominant energy range of the input signal with more discriminative features, which can be naturally and easily generalized and applied to different kinds of time series signals. The main contributions of this work are threefold:

- We perform *frequency spectrum energy analysis* for signal decomposition in *T-WaveNet*, wherein the frequency ranges with more energies are divided into finer-grained subbands and are thus represented with more dimensions compared with other low-energy frequency ranges in the feature vector, which facilitates effective learning of informative and discriminative features with a limited number of potentially noisy training samples.

- To extract the features effectively, we propose a novel invertible neural network (INN) based wavelet transform as the tree node in the *T-WaveNet*. Compared to fixed-basis wavelet *Haar* or lifting scheme-based wavelet, INN-based wavelet provides better representation capacity owing to its entirely data-driven characteristic. To the best of our knowledge, this is the first attempt to model wavelet transform using INN.

- Finally, inspired by the self-attention mechanism in Transformer, we propose an instrumental *feature fusion module* which considers the feature dependencies across different frequency components, thus effectively enhances the robustness of the model by mitigating the impact of the heterogeneity exists in sensor signals recorded from different subjects.

Extensive experiments on four popular sensor signal datasets, namely UCI-HAR for activity recognition, OPPORTUNITY for gesture recognition, BCICIV2a for intention recognition, and NinaPro DB1 for muscular movement recognition, show that our *T-WaveNet* consistently outperforms state-of-the-art solutions.

## 2   RELATED WORK

In this section, we review related work on feature extractions for time series analysis and wavelet transform modeling using deep learning techniques.

### 2.1   FEATURE EXTRACTION FOR SIGNAL ANALYSIS

Existing feature extraction methods for time-varying signal analysis can be broadly classified into hand-crafted and deep learning-based methods, and the former can be further divided into statistical and structural approaches.

Hand-crafted *statistical* and *structural* approaches are widely used in early studies. For instance, Kao et al. (2009) applies statistical features such as the mean and the mean absolute deviation (MAD) for online activity detection from a portable device. Duan et al. (2015) performs discrete wavelet transform on surface-Electromyography (sEMG) signal for hand motion classification. While relatively easy to calculate, these hand-crafted features are not effective for complicated tasks.

In recent years, DNNs have become the mainstream approaches for signal feature extraction. CNN-based models are widely used to extract local temporal correlations of time series data. For example, Lee et al. (2017) combines multiple CNN layers with different kernel sizes to obtain the temporal dependencies at various time scales. Lawhern et al. (2018), Lin et al. (2020), and Amin et al. (2019) utilize CNNs in BCI applications to establish end-to-end EEG decoding models and achieve promising performance. As CNN-based models are often insufficient for extracting long-term temporal features, Ordóñez & Roggen (2016), Yuki et al. (2018), and Chambers & Yoder (2020) propose to combine CNNs with LSTMs to extract both short- and long-term temporal features. Xu et al. (2019) bring out the Inception CNN structures to extract local temporal features at various time scales and utilize gated recurrent units (GRUs) to obtain global temporal representations. Recently, Transformer-based methods utilize the self-attention mechanism to model the global temporal dependencies and shows superior performance in various tasks. For example, Song et al. (2021) construct a simple yet effective Transformer-based model to obtain discriminative representations for EEG signal classification. Li et al. (2021) design a two-stream convolution-augmented transformer to extract features for human activity recognition, which considers both time-over-channel and channel-over-time dependencies.

The above deep learning based solutions try to extract features directly from the raw signal, which ignore the unique characteristics of each type of signals (e.g., the frequency spectrum information) and usually become less effective when training on scarce signal data. Notably, some works try to alleviate this issue by guiding the DNNs with the traditional time-frequency features such that more effective deep representations can be learned. For instance, Ito et al. (2018) feed the spectrogram images constructed from the temporal features of the inertial signals into CNN models to learn inter-modality features. Laput & Harrison (2019) train a CNN on the time-frequency-spectral features of the input sensor signal to obtain a fine-grained hand activity sensing system. Furthermore, S et al. (2019) integrates the Short-Time Fourier Transform into the DNNs to directly learn frequency-domain features. However, the above solutions often need to design tailored features for different kinds of signals, which is tedious and time-consuming, and severely limits the application scenarios.

### 2.2   WAVELET TRANSFORM MODELING

Frequency-domain feature extraction methods such as Fourier or Wavelet transform are more preferable to time-domain ones, since it is usually easier to extract discriminative features in the frequency domain than directly from the raw input signal. Compared with Fourier transform, which decomposes a signal into fixed frequency components, Wavelet transform is shown to have excellent properties for transient signal analysis, thanks to its capability to analyze signals at different frequencies with various time resolutions. In this section, we focus on Wavelet transform.

Recently, integrating wavelet transform with deep learning techniques is shown to be effective in signal and image processing. Early attempts directly replace certain layers in the DNNs with traditional wavelet transform to reduce the number of parameters as well as improve the interpretability. For instance, Williams & Li (2018) replace the max-pooling layer with a wavelet pooling algorithm to address the overfitting problem. Fujieda et al. (2017) propose to replace the pooling and convolution layers with wavelet transform.

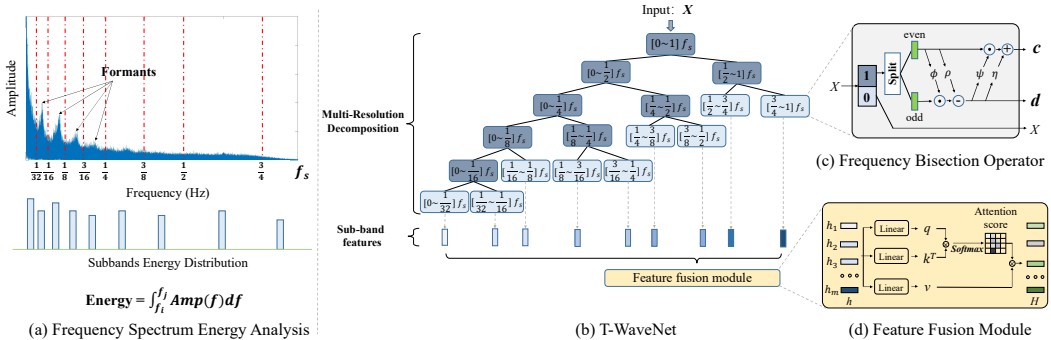

Figure 1: Given an input signal, we first perform **(a) Frequency Spectrum Energy Analysis** to decompose signal ($f_s$) into multiple frequency subbands with comparable energy, where each subband contains at most one formant. Then, according to the above decomposition, we construct **(b) T-WaveNet**, a tree-structured network, where each node is a **(c) Frequency Bisection Operator** built with an INN-based wavelet transform. The operator outputs the high- ($d$) and low-frequency ($c$) components of the input if its binary gate value is "1", and otherwise bypass the input ($\phi$, $\psi$, $\rho$, and $\eta$ have the same structure: Conv1D$(3, 1)$→LeakyRelu($\alpha = 0.01$)→Dropout($rate = 50\%$)→Conv1D$(3, 1)$→Tanh.). Considering the personalized heterogeneity of the input, we utilize a **(d) Feature Fusion Module** to fuse the subband features $\{h_i\}$ according to the feature dependencies across various frequency components. Finally, the enhanced feature vectors are supervised by Cross-Entropy loss for classification. See Section 3 for more details.

The above methods typically utilize a fixed wavelet basis (such as Haar (Haar, 1910)), which tend to be sub-optimal as it is less flexible and expressive for complex data. Recently, Rodriguez et al. (2020) propose to replace the fixed wavelet basis in the lifting scheme with deep learnable modules to realize an adaptive wavelet transform unit for image classification, aiming at learning more discriminative frequency features from images.

In this work, to better handle the complex spatial-temporal time series data, we build a novel deep wavelet transform unit with powerful representation capability, named *frequency bisection operator*. It decomposes the input signal into frequency subbands of various sizes and models the wavelet basis with an INN (Dinh et al., 2014). To the best of our knowledge, this is the first attempt to model the wavelet transform using INN. We detail the proposed solution in Section 3.2.

# 3 METHODS

Instead of solely relying on the neural network for information extraction from raw signals, *T-WaveNet* takes into consideration the uneven information distribution among different frequency ranges of signals and constructs a tree-structured network for feature learning, where those frequency ranges with more energy are divided into finer-grained subbands and represented with more dimensions in the feature vector, leading to more powerful representations embedding more desired information extracted from time series signals.

An overview of the *T-WaveNet* is shown in Fig. 1. We first perform *frequency spectrum energy analysis* (Fig. 1 (a)) to obtain the set of signal frequency subbands with various formants and energy envelops. Then, we construct *T-WaveNet*, a tree-structured network (Fig. 1 (b)), which iteratively decomposes the input signal into frequency subbands of various sizes according to *frequency spectrum energy analysis*. Each node on the tree is a *frequency bisection operator* built with an INN-based wavelet transform unit (Fig. 1 (c)). The operator is conditioned on a binary gate. It outputs the high-frequency and low-frequency components of the input if the gate value is "1", and otherwise bypass the input. In addition, different subjects may have some unique characteristics, resulting in distribution divergence. To deal with such individualized heterogeneity, after obtaining the set of features from each frequency subband, we utilize an effective *feature fusion module* to enhance the task-oriented subband features (Fig. 1 (d)).

### 3.1 FREQUENCY SPECTRUM ENERGY ANALYSIS

To quantitatively evaluate the information within the frequency range and perform the subband splitting accordingly for constructing the *T-WaveNet*, we perform *frequency spectrum energy analysis* (*FSEA*) to obtain a set of frequency subbands with roughly equivalent energy. In this way, the informative frequency range would be divided into finer subbands and be represented with more feature dimensions, easing the representation learning of signal data.

*FSEA* consists of two phases: a *formants-guidance frequency band splitting* phase to obtain the initial subbands set, and an *energy-guidance advanced subband splitting* to further balance the energy of each subband. In the first phase, given the input signal $X \in \mathbb{R}^N$, we first calculate the spectrum with the Fourier transform on the entire signal (Fig. 1 (a)) so as to obtain the overall frequency distribution of the signal. Next, we obtain the set of formants $\mathcal{P} = \{f_p\}_p$, where $f_p$ is a local maximum (*i.e.*, formant) of the envelope of spectrum[1]. Such formants represent the most direct source of the signal information. Then, we recursively bisect the frequency band until there is at most one formant that falls in each frequency subband $[f_i, f_j]$, where $f_i$ and $f_j$ are the starting and ending frequency, respectively. All these subbands are collected in set $\mathcal{Q}$. In the second phase, we calculate the energy of each subband in $\mathcal{Q}$ with Eq. (1) (Boashash, 2003) and further bisect the subband whose energy exceeds $\zeta$ times of $E_{\min}$, where $E_{\min}$ is the the minimum subband energy for subbands in $\mathcal{Q}$, and $\zeta$ is a scaling factor. This subband splitting phase further ensures that each resulted subband in the new set $\mathcal{Q}'$ contains similar amount of energy. Here, $Amp(f)$ is the amplitude of the frequency $f$.

$$Energy = \int_{f_i}^{f_j} Amp\left(f\right) d_f. \tag{1}$$

The *FSEA*, in principle, can be regarded as a perquisite for *T-WaveNet* construction. Based on $\mathcal{Q}'$ obtained from *FSEA*, we construct the proposed *T-WaveNet* in two steps: (i) *bottom-up* marking, and (ii) *pruning*. In the bottom-up marking process, given a full binary tree with some height (Fig. 1 (b)), we locate each subband in $\mathcal{Q}'$ and set its binary gate as "0", which serves as the leaves. Next, we set the binary gates of all other nodes on the path from the leaves to the root as "1". In the pruning process, the nodes without gate settings are removed from the tree. The leaves of resulted sub-tree cover all the frequency subbands in $\mathcal{Q}'$. With the above, *FSEA* divides the frequency ranges with more energy into finer-grained subbands.

### 3.2 FREQUENCY BISECTION OPERATOR

Each node in the *T-WaveNet* is a frequency bisection operator built with an INN-based second-generation wavelet transform. The second-generation wavelets, also known as *Lifting Scheme* theory (Sweldens, 1998), is a simple yet powerful approach to construct different wavelets such as *Haar*. The main idea is to utilize the strong correlation among the neighboring samples in the signal to separate the low-frequency (approximation) and high-frequency (details) subbands. The Lifting Scheme separates the input vector $X = (x[0], x[1], ..., x[2k - 1]), k \in \mathbb{N}$, into low- and high-frequency subbands in three steps.

- *Splitting*. The signal is split into two non-overlapping partitions as shown in Eq.(2). The general partition method is dividing the signal to even part $X_{even} = (x[0], x[2], ..., x[2k - 2])$ and odd part $X_{odd} = (x[1], x[3], ..., x[2k - 1])$. The spliting operater is:

- *Predictor*. The two partition sets are distributed alternatively in the original signal. Based on the signal correlation, it is possible to build a good predictor $P$ to predict one partition set from the other. One example to predict $X_{odd}$ from $X_{even}$ is shown in Eq. (3), where $d$ (details) denotes the difference between $X_{odd}$ and the prediction set $P(X_{even})$.

- *Updator*. As shown in Eq. (4), the details $d$ is further used to update the even part with an updator $U$ to preserve some consistent characteristics of the original signal, such as mean and higher moments. Here, $c$ is also called "approximation".

$(X_{even}, X_{odd}) = Splitting(X).$ (2)  $d = X_{odd} - P(X_{even}).$ (3)  $c = X_{even} + U(d).$ (4)

In traditional wavelet transform, the choice of the wavelet basis is important yet difficult, which would affect the analysis results significantly. Traditional wavelet construction framework with fixed

---

[1]In case that the signal is too noisy, one can first conduct standard denoising approaches in the corresponding domains to minimize the impact of noise.

coefficients $(P, U)$ usually lacks of adaptability to extract more informative features from complex signals. Therefore, we take advantage of the INN (Dinh et al., 2014), a bijective transformation, to build the wavelet transform unit, which can effectively model the correlations between the inputs and the outputs with learnable structures. To combine the Lifting Scheme with INN, the improvements include: (i) adapting Eq. $(3), (4)$ to an affine function, as shown in Eq. $(5), (6)$, which improves the transformation ability; (ii) learning the wavelet coefficients ($\phi$, $\psi$, $\rho$ and $\eta$) in Eq. $(5), (6)$ using separate CNN modules.

$$d = X_{odd} \odot \exp(\phi(X_{even})) - \rho(X_{even}), \quad (5); \quad c = X_{even} \odot \exp(\psi(d)) + \eta(d). \quad (6)$$

Here, $\exp(\phi(\cdot))$, $\exp(\psi(\cdot))$ stand for scale and $\rho$, $\eta$ stand for translation. $\odot$ is the element-wise production. The $\exp(\cdot)$ is to introduce non-linearity into the function and is omitted in Fig. 1 (c). Note that, if the values of $\phi$ and $\psi$ are set to zero, Eq. $(5), (6)$ will degenerate to the standard Lifting Scheme equations (Eq. $(3), (4)$).

The proposed INN-based wavelet transform has two advantages. First, being an invertible architecture, INN avoids information loss during feature extraction and, hence, inherits the conventional wavelet transform property. Second, the wavelet basis is entirely data-driven, and learning high-dimensional coefficients can be regarded as implementing various wavelet basis to the input simultaneously according to the requirements of the downstream task, which is more effective than conventional fixed-basis wavelets.

## 3.3 FEATURE FUSION MODULE

There inevitably exists personalized heterogeneity in the signal data collected on different subjects. Thus, the features extracted from a certain frequency subband may contribute differently to different subjects. To alleviate the effect of personalized heterogeneity and achieve better task performance, we build an efficient feature fusion module inspired by the well-known Transformer (Vaswani et al., 2017). The feature fusion module is built on the self-attention mechanism, which learns the personalized feature dependencies across different frequency subbands, and adaptively enhances the task-related subband features with higher weights during fusion, as illustrated in Fig. 1 (d).

To calculate the weights of the features $\mathbf{h} \in \mathbb{R}^{n \times m}$ extracted from the frequency subband set $\mathcal{Q}'$, we first calculate the query $\mathbf{q}$, keys $\mathbf{k}$, and values $\mathbf{v}$ from subband features $\mathbf{h}$:

$$\mathbf{q} = \mathbf{W}_q \mathbf{h} + \mathbf{b}_q, \mathbf{q} \in \mathbb{R}^{r \times m}; \quad \mathbf{k} = \mathbf{W}_k \mathbf{h} + \mathbf{b}_k, \mathbf{k} \in \mathbb{R}^{r \times m}; \quad \mathbf{v} = \mathbf{W}_v \mathbf{h} + \mathbf{b}_v, \mathbf{v} \in \mathbb{R}^{r \times m} \quad (7)$$

where $\mathbf{W}_* \in \mathbb{R}^{r \times n}$ and $\mathbf{b}_* \in \mathbb{R}^r$ are the weight matrix and bias, respectively, $m$ is the number of features, $n$ is the feature dimension of $\mathbf{h}$, and $r$ is the feature dimension of $\mathbf{q}, \mathbf{k}$ and $\mathbf{v}$. Then, we compute the attention weights for each value as the compatibility score of the query with Eq.(8). Then the output of the feature fusion module is calculated as the weighted sum of the values in Eq.(9).

$$\alpha = \text{softmax}(\mathbf{k}^T \mathbf{q}), \alpha \in \mathbb{R}^{m \times m}. \quad (8) \qquad \mathbf{H} = \alpha \cdot \mathbf{v}^T, \mathbf{H} \in \mathbb{R}^{m \times r}. \quad (9)$$

The flattened attentive frequency subband representation $\mathbf{H}_{\text{flatten}} \in \mathbb{R}^{(mr)}$ is finally fed into a standard softmax classifier to produce the prediction results:

$$\mathbf{p} = \text{softmax}(\mathbf{W}\mathbf{H}_{\text{flatten}} + \mathbf{b}), \mathbf{p} \in \mathbb{R}^C \quad (10)$$

where $\mathbf{W} \in \mathbb{R}^{C \times (mr)}$ and $\mathbf{b} \in \mathbb{R}^C$ are the weight matrix and bias, respectively, $\mathbf{p}_i$ is the $i$-th element of $\mathbf{p}$, which denotes the predicted probability for class $i = 1, \ldots, C$, and $C$ is the number of class.

## 3.4 LOSS FUNCTION

Our loss function is shown in Eq. $(11)$. The first term is the cross-entropy loss for classification, where $C$ denotes the number of classes, and $y_i$ is the binary ground-truth. Besides this term, we also explore incorporating an additional term to regularize the wavelet decomposition during training. The second loss term ensures that, for each frequency bisection operator, the mean value of the decomposition output $c_j$ is close to that of the input $x_j$. Here, $M$ is the total number of operators in the tree-structured network, and $\lambda$ tunes the strength of the regularization.

$$\mathcal{L} = -\sum_i^C y_i \log(\mathbf{p}_i) + \lambda \sum_j^M \|x_j - c_j\|_2. \quad (11)$$

Table 1: The four datasets used in our experiments.

| Datasets | # classes | # samples in the timing window | Signal types | # signals | Sampling rate(Hz) | # subjects | Data partitioning |
|---|---|---|---|---|---|---|---|
| OPPORTUNITY | 18 | 48 | Inertial data | 77 | 30 | 3 | Train: Subject 1:ADL(1,2,3,4,5), drill; Subject 2:ADL(1,2), drill; Subject 3:ADL(1,2), drill; Test: Subject 2:ADL(4,5); Subject 3:ADL(4,5) |
| UCI-HAR | 6 | 128 | Inertial data | 9 | 50 | 30 | Train/Test: 7:3 |
| BCICIV2a | 4 | 400 | EEG | 22 | 250 | 9 | Leave-one-subject-out |
| NinaPro DB1 | 52 | 150 | surface EMG | 10 | 100 | 27 | Training: Subjcets (1,3,4,6,7,8,9); Test: Subjects (2,5,10) |

# 4 EXPERIMENTS

In this section, we evaluate the performance of *T-WaveNet* on four datasets, namely OPPORTUNITY (OPPOR) (Ordóñez & Roggen, 2016), UCI-HAR (Davide et al., 2013), BCICIV2a (Brunner et al., 2008), and NinaPro DB1 (Manfredo et al., 2012). A brief description of these datasets is listed in Table 1. More details on experimental settings, additional experimental results and discussions (e.g., performance on other datasets, hyperparameter analysis, robustness, and computational time) are presented in the supplementary materials.

## 4.1 COMPARISON WITH STATE-OF-THE-ART METHODS

We compare *T-WaveNet* with state-of-the-art methods on various signal classification tasks and datasets, and the experimental results are shown in Tables 2-5.

As can be observed, the performances of previous state-of-the-art methods vary with different types of signals, while the proposed *T-WaveNet* consistently achieves better performance across all these datasets. We attribute it to the fact that the proposed method seamlessly and effectively integrates the inherent dominant energy range property of signals into our deep model. In the following, we present detailed analysis of comparison on benchmarks of each task.

Table 2: Performance comparison on UCI-HAR dataset.

| Methods | UCI-HAR | |
|---|---|---|
| | $Accuracy$ | $F_w$ |
| DeepConvLSTM | 0.908 | 0.905 |
| Res-LSTM | 0.916 | 0.915 |
| CNN-LSTM | 0.921 | - |
| InnoHAR | - | 0.945 |
| STFNet | 0.929 | 0.923 |
| Harmonic | 0.929 | 0.929 |
| LSTM-CNN | 0.958 | - |
| MI-CNN-GRU | 0.962 | 0.961 |
| *T-WaveNet* | **0.971 ± 0.012** | **0.974 ± 0.009** |

Table 3: Performance comparison on OPPOR dataset.

| Methods | OPPOR | |
|---|---|---|
| | $F_m$ | $F_w$ |
| DeepConvLSTM | 0.672 | 0.915 |
| LSTM-S | 0.698 | 0.912 |
| LSTM Ensembles | 0.726 | - |
| ARN | - | 0.903 |
| Res-LSTM | - | 0.905 |
| Harmonic | 0.575 | 0.894 |
| FilterNet | 0.743 | 0.928 |
| *T-WaveNet* | **0.763 ± 0.011** | **0.931 ± 0.013** |

**Human activity recognition (UCI-HAR and OPPOR)**

As can be observed from Table 2, *T-WaveNet* outperforms both the recent end-to-end deep models (Harmonic (Hu et al., 2020) , LSTM-CNN (Xia et al., 2020), MI-CNN-GRU (Dua et al., 2021)) and a hybrid model (STFNet (S et al., 2019)) on the UCI-HAR dataset. To be specific, compared with *MI-CNN-GRU*, *T-WaveNet* yields 2.18% and 1.35% relative improvements in terms of $Accuracy$ and $F_w$ score, respectively. This is because, *MI-CNN-GRU* solely relies on combining the CNN and GRU model for feature extraction, without considering the essential information in the frequency domain. In contrast, *STFNet* integrates the STFT (Short-Time Fourier Transform) into the neural network to learn features directly in the frequency domain. However, it does not distinguish the roles of various frequency components. *T-WaveNet* decomposes the most informative frequency range into finer-grained subbands, which increases the representation capacity for feature learning under limited training samples. Similarly, for the OPPOR dataset in Table 3, compared with the popular CNN-LSTM structure used in *FilterNet* (Chambers & Yoder, 2020) , *T-WaveNet* clearly demonstrates better feature learning capability with a 2.66% improvement in $F_m$ score. As for, and 2.66% improvement of $F_m$ for OPPOR. Such improvements clearly demonstrate the feature learning ability

of the proposed method. Moreover, *T-WaveNet* can better capture the inter-class differences than existing solutions, because it learns the feature dependencies across different frequency subbands.

**Brain intention recognition (BCICIV2a)**

It is challenging to learn effective features for brain intention recognition since EEG signals are usually quite noisy. As shown in Table 4, *T-WaveNet* achieves superior performance on BCICIV2a compared with standard CNN classifiers (e.g., CTCNN (Schirrmeister et al., 2017) and EEGNet (Lawhern et al., 2018)). The *EEG-Image* (Bashivan et al., 2018) model tries to apply all sorts of hand-crafted features (i.e., spectral, spatial, and temporal features) into the *CNN-RNN* architecture, but it actually mislead the neural network in feature learning, resulting in poor performance. The graph neural network-based classifier *NG-CRAM* (Zhang et al., 2020) is effective in modeling the spatial correlations among signals, but it also suffers from noises during message passing among neighbouring nodes. Finally, *MIN2Net* (Autthasan et al., 2021) utilizes auto-encoder to extract features from the input signals, but it may lose some essential information during this procedure. Also, the implementation of the Conv2D may restrict the model's ability in learning long-term temporal dependencies. In contrast, *T-WaveNet* concentrates on the features extracted from the dominant energy range with more discriminative power, and the proposed INN-based wavelet can avoid the information loss during feature extraction, thereby yielding more robust representations.

Table 4: Performance comparison on BCI-CIV2a dataset.

| Methods | BCICIV2a |
|---|---|
| | *Accuracy* |
| CTCNN | $0.4767 \pm 0.1506$ |
| EEGNet | $0.5130 \pm 0.0518$ |
| EEG Image | $0.3270 \pm 0.0430$ |
| Cascade model | $0.3183 \pm 0.0399$ |
| Parallel model | $0.3267 \pm 0.4499$ |
| NG-CRAM | $0.6011 \pm 0.0996$ |
| MIN2Net | $0.6033 \pm 0.0924$ |
| *T-WaveNet* | $\mathbf{0.6301 \pm 0.0212}$ |

Table 5: Performance comparison on NinaPro dataset.

| Methods | NinaPro |
|---|---|
| | *Accuracy* |
| GengNet | 0.778 |
| MV-CNN | 0.874 |
| TCN | 0.898 |
| HuNet | 0.870 |
| WeiNet | 0.850 |
| XceptionTime | 0.918 |
| DLPR | 0.911 |
| *T-WaveNet* | $\mathbf{0.932 \pm 0.0103}$ |

**Muscular movement recognition (NinaPro DB1)**

As shown in Table 5, *T-WaveNet* achieves a relative $6.65\%$ improvement over *WeiNet* (Wei et al., 2019) that trains a CNN classifier on hand-crafted features, and $1.52\%$ improvement over *XceptionTime* (Rahimian et al., 2020) that solely relies on time domain features. *DLPR* (Pancholi et al., 2021) combines both the time- (peaks and zero-crossing) and frequency- (power spectral) domain features with CNNs and achieves remarkable performance for this task. However, this solution requires lots of domain knowledge and much efforts to design signal-specific features, which is challenging to be generalized to other signal types. In contrast, the proposed INN-based wavelet transform in *T-WaveNet* is adaptive for learning information from both time- and frequency-domain, which facilitates to extract more discriminative features for muscular movement recognition.

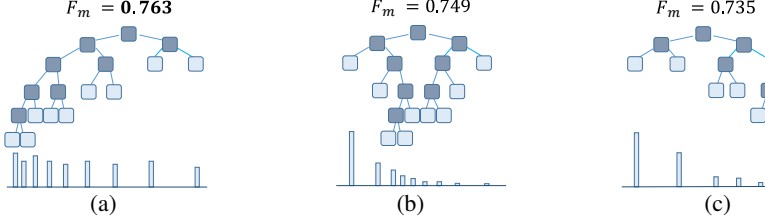

Figure 2: **Effectiveness of FSEA for tree structure configuration.** The histograms below denote the energy preserved in the frequency subband of each leaf. (a) is the configuration obtained with FSEA (Section 3.1), wherein the energy contained in each subband is similar. (b) and (c) are other two exemplar configurations with uneven energy distribution due to the coarser split of the dominant frequency range, all of which suffer from performance degradation (decreased $F_m$). See Section 4.2.

## 4.2 ABLATION STUDY

Here, we study the impact of each component in *T-WaveNet* by performing a detailed ablation study on the challenging OPPOR dataset and the corresponding frequency spectrum is shown in Fig.1 (a).

**Impact of frequency spectrum energy analysis**. To verify the effectiveness of *FSEA*, we train two other variants of the *T-WaveNet* with different feature vector distributions (Fig. 2). Fig. 2 (a) shows that the tree structure constructed with the dominant energy range concept outperforms other variants, indicating that dominant frequency bands represented with more dimensions in feature vectors indeed contain more discriminative information for classification. The impact of the two-phases subband splitting scheme is shown in the supplementary.

**Impact of frequency bisection operator**. To verify the effectiveness of our INN-based wavelet transform in the frequency bisection operator, we also experiment with two variants: (i) *T-WaveNet-Haar*, which replaces the INN-based wavelet transform with the traditional Haar wavelet basis; (ii) *T-WaveNet-LS*, which implements the deep version of the Lifting Scheme, in which the $P$ and $U$ in Eq.(3)(4) are realized using the same deep modules as $\phi$, $\psi$, $\rho$ and $\eta$ (Fig.1 (c)). The results in Table 6 show that our *T-WaveNet* achieves $18.6\%$ and $3.8\%$ improvements on $F_m$ score in OPPOR dataset over *T-WaveNet-Haar* and *T-WaveNet-LS*, respectively. We attribute the improvements to the fact that our high-dimensional deep wavelet coefficients are learned from data. Thus, our INN-based wavelet is more adaptable to various signals compared with the fixed Haar wavelets, and has higher representation capacity than the deep Lifting Scheme. See the supplementary for more details.

**Feature fusion module**. To demonstrate the effectiveness of the proposed adaptive feature fusion module for handling the personalized heterogeneity of the signal data, we remove the fusion module from *T-WaveNet* and fuse the leaf features $\{\mathbf{h}_i\}$ with equivalent weights (*T-WaveNet-noFusion*). As shown in Table 6, the performance of the resulted model decreases by $2\%$-$3\%$, which indicates that the proposed feature fusion module learns the feature dependencies across various frequency components and enhances task-related features for classification. More details are presented in the supplementary.

Table 6: Results with structural variants. "$-Haar$" means the frequency bisection operator is replaced by the traditional Haar wavelet. "$-LS$" denotes the deep version of the Lifting Scheme. "$-noFusion$" means the feature fusion module is removed from *T-WaveNet*.

| Structural variants | $Fm$ | $Fw$ |
|---|---|---|
| *T-WaveNet-Haar* | $0.644 \pm 0.012$ | $0.908 \pm 0.016$ |
| *T-WaveNet-LS* | $0.735 \pm 0.008$ | $0.926 \pm 0.007$ |
| *T-WaveNet-noFusion* | $0.747 \pm 0.013$ | $0.928 \pm 0.006$ |
| *T-WaveNet* | $\mathbf{0.763 \pm 0.011}$ | $\mathbf{0.931 \pm 0.013}$ |

## 4.3 DISSCUSSION AND LIMITATION

Our approach is currently more suitable for the time-series data with some specific patterns in their frequency spectrum across subjects/cases. Biological signals often have such characteristics and in this regard, we extensively evaluate the proposed network on biological signals. However, if other time-series signals have similar properties, our network can also been applied to analyze them and achieves performance gains (see the supplementary for more experimental results). On the contrary, if the time-series data does not have this characteristic, especially when the frequency energy spectrum is time-variant (e.g., some financial data), the proposed network cannot achieve remarkable performance improvements.

## 5 CONCLUSION

We propose *T-WaveNet*, a novel tree-structured wavelet neural network for time series signal analysis, which exploits an inherent property of various types of signals, i.e., the *dominant frequency range*, to more efficiently extract informative representations from input signals than most existing solutions that depend only on the raw data. We first conduct frequency spectrum energy analysis of the signals to get a set of dominant frequency subbands. We then construct a tree-structured network with INN-based wavelet transform units to iteratively decomposes the input signal into various frequency subbands with similar energies. Finally, all the leaf node features are adaptively fused to mitigate the impact of the individualized heterogeneity of the signals. Experimental results on four datasets clearly show the superiority of *T-WaveNet* over other state-of-the-art solutions for time series classification task. This work, in general, offers insights on how to leverage both data and empirical knowledge to drive a learning model towards more powerful representations for time series signals, which is of great significance for many practical applications with a limited amount of training data.

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
