# OpenReview forum: "T-WaveNet: A Tree-Structured Wavelet Neural Network for Time Series Signal Analysis"
_ICLR.cc/2022/Conference — ICLR 2022 Poster_

### Official Review · Reviewer_4oWV · 2021-10-24

**Correctness:** 3
**Technical Novelty And Significance:** 3
**Empirical Novelty And Significance:** 3
**Recommendation:** 6
**Confidence:** 4

**Main Review:**

In general, the paper gives a reasonable way to combine classic frequent-temporal analysis (i.e., Fourier transformation and wavelet) and deep learning operations (e.g., attention for feature fusion). I list the main advantages and disadvantages below:

Advantages:
+  The combination of deep learning and classic frequent-temporal analysis is effective, e.g., to use attention based mechanism for feature fusion across different domain is a solid design.
+ The organization of empirical study is good, the author not only reports the end-to-end performance boost over the state-of-the-art approaches, but also provides some ablation study to support the design of the wavenet modules.

Disadvantages:
+ The biggest concern I have is that, the author claims "Such a disentangled representation learning method facilitates a more effective extraction of the discriminative features, as demonstrated by our extensive experiments on various real-life time series classification datasets." Along with the tune of the introduction, it seems that the author tends to argue this approach should be considered as a general approach for various time series analysis tasks. However, in the empirical study only four classification tasks over human activity or health monitoring data are considered. I think such empirical finding may not be sufficient for a claim of an effective general-purpose time series analysis framework.

+ One technique section is a little confusing to me, i.e., why would the FSEA balance the energy of each subband? It seems that the motivation of such frequent domain analysis is to identify the dominating frequency and remove the noisy in other frequencies. Additionally, shouldn't the wavelet transform include the functionality of co-locating salient frequent and temporal signal simultaneously? Why would you still need a FSEA module?

+ Lastly, a tiny issue would be a lack of introduction about how the training and test datasets are split in the text, perhaps one additional col in Table 1 would fix this.



**Summary Of The Paper:**

This paper introduces T-Wavenet, a fused approach including both classic feature engineering and recent advance of deep learning, in an attempt to improve the state-of-the-art analysis of time series. Empirical study is mainly conducted on human activity or health monitoring data to support the design decisions.

**Summary Of The Review:**

This paper presents a solid and reasonable way to combine classic frequent-temporal analysis and the recent advance of deep learning based approaches.
However, it seems a little incautious to claim the effectiveness of the proposed model as a general purpose approach for time series analysis based on the narrow scope of the benchmark datasets. I personally suggest two options for fixing this:

+ To conduct a more extensive evaluation over diversified datasets, e.g. there are 128 datasets available in the latest UCR time series website (https://www.cs.ucr.edu/~eamonn/time_series_data_2018/), all of them or a representative subset of them would be a good collection of benchmarks.

+ To narrow-down the scope of the approach as an effective method for health monitoring related time series analysis. In this way, some concrete analysis should be added about some unique domain-specific properties of such signals, and the author would also justify why the proposed approach would be able to learn the representation of these properties more effectively.

Post rebuttal comments:
My concerns have been mostly resolved given the update and additional experimental results.

---

### Official Review · Reviewer_jwqd · 2021-11-01

**Correctness:** 3
**Technical Novelty And Significance:** 3
**Empirical Novelty And Significance:** Not applicable
**Recommendation:** 6
**Confidence:** 3

**Main Review:**

This is quite an interesting approach for the real world time series signal analyses, and seems promising in terms of accuracy.

Major point.
One thing missing is an experiment for the authors' argument about an advantage on a limited amount of training data. This point, the robustness and performance of the T-WaveNet, should be proofed by experiments where training data size is gradually decreasing (75%, 50%, 25%, etc.).

Minor point.
What is the unit of "Timing windos size" in the table 1?

**Summary Of The Paper:**

The authors proposed T-WaveNet, a novel tree-structured wavelet neural network for time series signal analysis. It utilizes the dominant frequency range to extract informative representation from raw signals. The tree-structure network consists of invertible neural networks as a frequency bisection operator, and of a feature fusion module. T-WaveNet was tested against various other methods using 4 different datasets and showed better results consistently.

**Summary Of The Review:**

T-WaveNet, a novel tree-structured wavelet neural network is proposed for real world time series signal analysis. If the authors could show robustness of the method against a limited amount of training data by experiment, this paper would be valuable for many participants of ICLR.

---

### Official Review · Reviewer_jjB7 · 2021-11-01

**Correctness:** 3
**Technical Novelty And Significance:** 3
**Empirical Novelty And Significance:** 3
**Recommendation:** 6
**Confidence:** 2

**Main Review:**

Pros.

-The well-founded idea combines frequency-based features (hand-imposed representation) and deep learning enhancement for time-series discrimination.
-As a result, a data-driven wavelet approach is obtained using INN schemes.
-Overall, the paper is clear and easy to follow. Moreover, the experiments dñemonstrate convincing evidence regarding the time-series discrimination results.
-The codes are available, and the main training details are exposed.

Cons. and comments

-The formant-based feature extraction seems to be confusing. How do you test the formant (maximum) estimation from noisy signals?.
-Authors claim that a fixed window is used to compute de FFT; however, how to set such a parameter for different non-stationary patterns?
-The loss function in eq 11 depicted an L2 regularization; how to deal with outliers?
-The bisection operation includes the estimation of even and odd signal parts. Nonetheless, It would be helpful to discuss better the relationship between the sampling frequency, the window size, and the bisection operation.
-Concerning the BCI2a database results, the acc seems to be poor. Can you explain better why?. Maybe, confusion matrices and class-dependent time-series analysis could be discussed.

**Summary Of The Paper:**

The authors introduce a deep learning approach to classify time series. In particular, a wavelet-based algorithm is carried out using invertible neural networks and a formant-based strategy. Besides, self-attention mechanisms are utilized to code short and long time-series correlations. Finally, different databases are tested, showing an interesting performance for non-stationary pattern coding.

**Summary Of The Review:**

A good paper is presented founded on a deep learning-based representation strategy that can be used as a data-driven wavelet representation holding attention mechanism. The experiments show the potential impact in different databases devoted to time-series classification.

---

### Official Review · Reviewer_VTwR · 2021-11-02

**Correctness:** 4
**Technical Novelty And Significance:** 4
**Empirical Novelty And Significance:** 3
**Recommendation:** 8
**Confidence:** 3

**Main Review:**

The paper proposes a truly innovative technique, which seems to be effective and versatile enough to be applied on several tasks and different biomedical signals (EEG, EMG,…)
The technique has been thoroughly evaluated and the results are convincing. The sue of the ablation study demonstrates the right decisions and design choices have been taken throughout the architecture design.
I have some minor points to raise:
In section 3.2, the frequency bisection operator aims at building a predictor for estimation odds samples from even ones. If I am not mistaking, this predictor is applied also for downsampled signals, but I fail to understand how hard it is to find a suitable predictor. How can the authors explain, why trying to achieve such a trivial task allows to build such an effective network.
Could the authors discuss how the use of scattering network with the suggested fusion module could be working on the different teste tasks?
Could the authors discuss how much computing time is required for training and deploying the suggested technique on real examples?


**Summary Of The Paper:**

This paper introduces a novel technique for the analysis of biomedical signals. The technique is inspired from Wavelet transform and consists in analyzing the original signal in different frequency subband. The technique differs from wavelet transform by the fact that instead of using a fixe wavelet function, a dedicated invertible neural network is learnt for each frequency subband. Finally, features from each subbands are fused using a Transformer like model.
The proposed techniques has been thoroughly tested on separate datasets and separate tasks, and compared to various state-of-the-art techniques, and the proposed technique has consistently outperformed other techniques on these various tasks. Finally, an ablation study has been performed in order to demonstrate the benefits of the different choices of the model architecture.


**Summary Of The Review:**

The authors suggested an innovative solution for the analysis of biomedical time-series. They have performed thorough testing of their approach on several tasks, and obtained convincing results.

---

### Decision · Program_Chairs · 2022-01-20

**Decision:**

Accept (Poster)

**Comment:**

This paper introduces a tree-structured wavelet deep neural network to effectively extract more discriminative and expressive feature representations in time series signals.  Based on a frequency spectrum energy analysis,  the approach decomposes input signals into multiple subbands and builds a tree structure with data-driven wavelet transforms the bases of which are learned using invertible neural networks. In the end, the scattering subband features are fused using a self-attention-like mechanism.  The effectiveness of the proposed approach is verified extensively on a variety of datasets from different domains including follow-up experiments in the rebuttal.  Overall,  the work is technically novel and provides an interesting way of extracting adaptive finer-grained features to deal with time series signals.  The authors' rebuttal is solid which has cleared most of the concerns raised by the reviewers with additional supportive experimental evidence.

I would recommend accept.